# Effect of 4 Weeks of High-Intensity Interval Training (HIIT) on VO_2_max, Anaerobic Power, and Specific Performance in Cyclists with Cerebral Palsy

**DOI:** 10.3390/jfmk10020102

**Published:** 2025-03-24

**Authors:** Cristian A. Lasso-Quilindo, Luz M. Chalapud-Narvaez, Diego C. Garcia-Chaves, Carlos Cristi-Montero, Rodrigo Yañez-Sepulveda

**Affiliations:** 1Group GIICSH, Faculty of Social Sciences and Humanities, Corporacion Universitaria Autonoma del Cauca, Popayan 190001, Colombia; luzchalapud@gmail.com; 2Research Group Applied Studies in Sport, INTEMED Research Incubator, Faculty of Education and Sport Sciences, Institucion Universitaria Escuela Nacional del Deporte, Cali 760041, Colombia; diego.garcia@endeporte.edu.co; 3IRyS Group, Physical Education School, Pontificia Universidad Catolica de Valparaiso, Valparaiso 2362807, Chile; carlos.cristi.montero@gmail.com; 4Faculty of Education and Social Sciences, Universidad Andres Bello, Viña del Mar 8370134, Chile; rodrigo.yanez.s@unab.cl

**Keywords:** para-sport, paralympic sport, HIIT, cycling athletes, sports performance

## Abstract

Background: High-intensity interval training (HIIT) is an effective and efficient method for training Paralympic athletes with cerebral palsy, particularly in intermittent sports and those requiring aerobic and anaerobic capacity, speed, and strength to delay fatigue onset and optimize athletic performance. Objectives: This study aimed to analyze the effects of four weeks of HIIT on the estimated VO_2_max, anaerobic power, and athletic performance in cyclists with cerebral palsy. Materials and Methods: This quasi-experimental study included three male athletes (Athletes A, B, and C) with cerebral palsy from the Paracycling Departmental Commission of Cauca, Colombia. The estimated VO_2_max was assessed using an incremental test on a cycling ergometer. Anaerobic power was measured using the 30 s long Wingate Anaerobic Test (WAnT_30 s. Specific performance was evaluated with an individual time trial of 14 km for class T1 and 20 km for class T2. HIIT training was performed on a cycling ergometer over four weeks (two sessions per week). The training intensity was based on watts (W) measured in the incremental test for long HIIT sessions and in the WAnT_30 s test for short HIIT sessions. The training load was monitored through heart rate (HR) responses and the subjective perceived exertion (RPE) at the end of the training. Results: After the HIIT intervention, percentage changes in the estimated VO_2_max were observed in Athlete A (+7%) and Athlete C (+9.4%). In the WAnT_30 s, there were increases in the maximal and mean power in Athlete A (>31%, 282.3 vs. 370.4 W), Athlete B (>15%, 272.5 vs. 312.6 W), and Athlete C (>9%, 473.7 vs. 516.2 W). Individual time trial performance improved, with reduced completion times for Athlete A (−6.7%, 2492 vs. 2325 s), Athlete B (−3.7%, 2486 vs. 2390 s), and Athlete C (−3.7%, 2775 vs. 2674 s). Conclusions: This study found that eight sessions of high-intensity interval training (HIIT) over a four-week period had a positive effect on the estimated VO_2_max in Athletes A and C. Moreover, all three paracyclists demonstrated improvements in their maximal and average power output during the 30 s Wingate Anaerobic Test (WAnT_30 s), as well as enhanced performance in the time trial test.

## 1. Introduction

Paracycling is the third most prominent Paralympic sport, conducted under the parameters of the International Cycling Union (UCI) and the International Paralympic Committee (IPC) [1,2]. The IPC and UCI oversee the functional classification and official competitions for male and female athletes with physical disabilities, visual impairments, and cerebral palsy [3,4].

Cerebral palsy is not a disease but rather an umbrella term that encompasses a variety of motor disorders of diverse origins, which may change with age [5]. It is defined as a group of permanent, yet not unchangeable, impairments in movement and motor function caused by a lesion, anomaly, or non-progressive alteration in the developing brain [6]. These disorders manifest in infancy and persist throughout life, affecting posture and motor control. While cerebral palsy itself does not worsen over time, its manifestations can evolve with growth [7]. In addition to motor difficulties, individuals with cerebral palsy commonly experience sensory, perceptual, and cognitive impairments, as well as epilepsy and associated musculoskeletal disorders [8].

This sport is performed through track races in a velodrome and road events on closed circuits [9]. Track events include time trials, individual pursuits, and team sprints. Road events consist of mixed team relay races and time trials. In Paralympic sports, functional classification ensures that athletic achievement is determined by factors other than the impact of disability, fostering equity in competition [10]. Functional classification in paracycling assigns athletes to classes for bicycles (C1–C5), handcycles (H1–H5), tandems (B), and tricycles (T1–T2) [11,12].

In the T1–T2 sport class, athletes are diagnosed with cerebral palsy, including hypertonia, ataxia, and athetosis [9,13]. They compete using a modified three-wheeled bicycle, with two rear wheels and one front wheel [14]. This configuration enhances stability and helps prevent accidents during training and competitions due to the instability of the Paralympic athlete [15,16]. Tricycle events range from 15 km to 20 km in individual time trials and from 30 km to 40 km in road races [4,9].

In paracycling, energy demands during events are primarily met by the oxidative pathway throughout most of the race, with a significant contribution from the glycolytic pathway during inclines and at the end of the race with a sprint [17,18]. This sport requires high lower-limb muscular power output to complete road and time trial races in the shortest possible time [19]. Thus, monitoring and controlling these variables is essential to meet the physiological and neuromuscular demands of competition [20].

Given this, coaches aim to enhance Paralympic athlete performance throughout the season, emphasizing sport and athlete characterization, physiological load monitoring, intensity and volume control, physical fitness evaluation, and competition performance assessment. These factors are vital in Paralympic sports due to the population’s variability and heterogeneity [1].

Performance in paracycling largely depends on the aerobic and anaerobic power generated during competition. These capacities should be trained using methods that ensure optimal physiological and competitive adaptations [21,22]. Research has shown that one of the most time-efficient training methods applicable to athletes with physical disabilities, visual impairments, or cerebral palsy is high-intensity interval training (HIIT).

For instance, a review conducted by Lasso-Quilindo and Chalapud-Narváez [23] analyzed multiple studies and found that this training method produces significant short-term improvements in aerobic and anaerobic capacity, body composition, and muscle fiber recruitment. Additionally, they observed that the effects are highly significant when HIIT is combined with circuit training, strength training, Pilates, and sprint exercises. Therefore, HIIT is considered an effective and efficient training strategy for Paralympic athletes with physical, visual, and cerebral palsy-related impairments, particularly in intermittent sports and disciplines that require aerobic and anaerobic capacity, speed, and strength to delay fatigue onset and optimize athletic performance.

Numerous studies in wheelchair basketball [24,25], wheelchair rugby [26], cerebral palsy football [27], paratriathlon [28], paracycling [20,29], and para-athletics [30] have reported changes in the maximum oxygen consumption (VO_2_max), upper- and lower-body muscular power, body composition, and athletic performance in Paralympic athletes [23].

HIIT consists of performing repeated bouts of high-intensity efforts, typically above 85% of Maximum Oxygen Uptake (VO_2_max), interspersed with active or passive recovery periods at low intensity [30,31]. In this endurance training method, the workload is characterized by a low training volume and high intensity (above 85% VO_2_max) during the athletic season [32]. This feature of HIIT is generally used to improve cardiorespiratory function and the energy reserves of oxidative and glycolytic metabolism, which translates into performance improvements in time trial sports by delaying the onset of fatigue [33].

To achieve the desired central and peripheral physiological adaptations, the dosing of HIIT workload is influenced by the stage of the season and the type of HIIT applied (short HIIT or long HIIT). Long HIIT intensities range from 2 to 6 min (85–100% VO_2_max), focusing primarily on the oxidative pathway. In contrast, short HIIT, which predominantly involves the glycolytic pathway, ranges from 10 to 60 s (100–115% VO_2_max), alternating with recovery intervals at 50–60% VO_2_max [33].

The study of HIIT in paracycling is crucial for optimizing athletic performance, as it enhances VO_2_max, anaerobic power, and muscular endurance—key factors in road and time trial competitions. Additionally, it allows training to be tailored to the specific needs of each functional category (bicycle, handcycle, tandem, and tricycle), promoting targeted physiological adaptations while reducing the risk of injury. Its impact on energy efficiency improves effort tolerance and delays fatigue. From a scientific perspective, the limited number of studies in this area restricts evidence-based sports planning. Finally, HIIT not only enhances performance but also provides significant health benefits, helping prevent comorbidities associated with disability.

Previous research on Paralympic sports performance has highlighted the need for more specific studies on elite athletes with disabilities [34]. A literature review on the use of HIIT reported studies on paracycling in the cycling and handcycling sport classes, but no studies were found for tricycles [23].

However, to date, these studies have primarily focused on measuring performance in controlled environments, improving mobility, and rehabilitation to avoid exacerbating athletes’ limitations [35,36]. Therefore, this study aims to analyze the effects on the estimated VO_2_max, anaerobic power, and athletic performance of cyclists with cerebral palsy after four weeks of HIIT.

## 2. Materials and Methods

### 2.1. Study Design

This quasi-experimental, longitudinal analytical study was conducted with Paralympic athletes from the Departmental Paracycling Commission of Cauca, Colombia. The dependent variables were aerobic power, anaerobic power, and specific performance. The independent variable was the four-week HIIT training program.

This study was approved by the Ethics Committee of the Vice-Rectory for Research at the University of Cauca and the Autonomous University Corporation of Cauca through Resolutions No. 011 of 2023 and No. 0221 of 2024, respectively. All procedures followed ethical guidelines in accordance with the Declaration of Helsinki, Resolution 8430 of 1993 issued by the Colombian Ministry of Health, and Data Protection Law 1581 of 2012.

### 2.2. Participants

This study involved three male Paralympic athletes (Athletes A, B, and C): Athlete A (years of disability: congenital; sport class: T1; age: 19 years; height: 156 cm; weight: 54.8 kg; competitive experience: 5 years), Athlete B (years of disability: congenital; sport class: T1; age: 24 years; height: 161 cm; weight: 51.4 kg; competitive experience: 4.5 years), and Athlete C (years of disability: 9 years; height: 171 cm; weight: 72.4 kg; sport class: T2; age: 56 years; competitive experience: 8 years). They actively participate in regional and national competitions (Qualification Games and National Paralympic Games), with a weekly training frequency ranging from 3 to 7 sessions.

The sample was non-probabilistic and intentional, with exclusion criteria including musculoskeletal injuries or inability to complete this study. Participants signed informed consent forms before starting this study. They were informed of the purpose of this research, were familiarized with the procedures, risks, and benefits, and could withdraw from this research at any time without any loss.

### 2.3. Procedure

This study was conducted over a six-week period. The first week was used for preintervention evaluations; from the second to the fifth week, the HIIT training was implemented; and the sixth week was dedicated to final evaluations.

### 2.4. Reference Tests

Measurements were taken before and after the four-week intervention during the first and last weeks of this study under the same conditions of place and time. Tests were conducted on three separate days with a 48 h interval between tests to avoid fatigue. Participants followed their usual diet, ensured proper hydration, refrained from high-intensity activities 48 h before the test, and abstained from consuming caffeine, supplements, and stimulants on test mornings.

Before starting protocols (initial and final tests) and training (during the intervention) on the Cyclus 2 bicycle ergometer (Avantronic, Cyclus 2^®^, RBM elektronik-automation GmbH, Leipzig, Germany), the device was calibrated according to the manufacturer’s instructions. The Paralympic athletes performed a standardized 10 min warm-up at a self-selected cadence of 60 revolutions per minute (rpm) with a load of 60 watts (W). Cool-down consisted of 5 min at 50 W and a cadence of 60 rpm.

### 2.5. Body Composition

To minimize potential measurement errors associated with bioelectrical impedance analysis (BIA), the Paralympic athletes ensured appropriate fluid voiding prior to assessment. Height was measured using a mechanical wall-mounted stadiometer (Seca 206 body meter GmbH & Co. KG., Hamburg, Germany). Body composition was assessed via BIA using a validated bioelectrical impedance scale (OMRON Healthcare Technology Co., Ltd., HBF-514C^®^, Kyoto, Japan), which has demonstrated high validity (*r* = 0.942) and reliability (ICC = 0.933–0.993) and has been previously employed in similar research contexts [30,37,38,39]. The device was calibrated according to the manufacturer’s instructions. The device sends a light electrical current (50 kHz) through the body via electrodes. Participants stood upright, wearing minimal clothing and no shoes, with their feet on the platform electrodes and hands gripping the designated handles, extending their arms forward at chest height.

### 2.6. Estimated VO_2_max

The estimated VO_2_max was indirectly measured using a graded, incremental test, where athletes were required to sustain increasing workloads over time. The test respected the principle of individuality, with specific adjustments for each athlete. For the T1 athletes, the initial workload was set at 70 W, with increments of 15 W every 2 min at a cadence of 60 rpm until exhaustion. For the T2 athlete, the initial workload was 100 W, increasing by 25 W every 2 min at a cadence of 80 rpm until exhaustion [20,40].

The test concluded when participants could no longer maintain the workload or cadence. In this test, heart rate (HR) was monitored each time the athletes progressed to a higher stage, as well as their maximum HR (*HR*_max_). The device used was a Polar H10 chest strap sensor (Polar Electro Oy^®^, Kempele, Finland) linked to the Cyclus 2 (Avantronic, Cyclus 2 ^®^, RBM elektronik-automation GmbH, Leipzig, Germany). Finally, it is important to consider the maximum watts (*W*_max_), body weight, and age to indirectly estimate the VO_2_max (mL·kg^−1^·min^−1^), using the following equation, which has been applied in previous studies [41,42]:Estimated VO_2_max = (10.51 × *W*_max_) + (6.35 × *Weight* (kg)) − (10.49 × *Age* (*years*)) + 519.3.

Although this equation has been applied in athletes without disabilities, previous studies have highlighted the need for prediction or estimation models of VO_2_max in sports that demand high aerobic capacity [43], such as paracycling in the T1 and T2 sport classes.

### 2.7. The 30 s Long Wingate Anaerobic Test (WAnT_30 s)

The measurement of anaerobic power was conducted using a Cyclus 2 bicycle ergometer (Avantronic, Cyclus 2^®^, RBM elektronik-automation GmbH, Leipzig, Germany). The test required the athlete to pedal against a resistance of 0.075 kg per kilogram of body mass at maximum speed for 30 s to determine the peak power output in watts [44]. Participants were instructed to reach their maximum power in the shortest time possible and maintain maximum power until the end of the test and were provided with auditory stimuli to encourage maximum effort [45].

The workload for the T1-class athletes was set at an initial cadence of 70 rpm, while for the T2-class athlete, the starting cadence was 90 rpm. The protocol began with a starting signal, where athletes were positioned standing on the pedals, with their dominant leg ready for the initial push. Athletes performed three alternating attempts, with 30 min of recovery between each attempt, and the best result was recorded.

### 2.8. Specific Performance

The performance test was conducted on a closed 1 km circuit. Athletes performed a standardized 20 min warm-up on the track at a comfortable pace, allowing them to familiarize themselves with the circuit. Timing was recorded using Witty·Gate photocells (Microgate, Bolzano, Italy) with a precision of ±0.4 milliseconds, positioned 40 cm above the ground and spanning a length of 2 m. For this test, the Paralympic athletes were instructed to cover the assigned distance in the shortest possible time. A 14 km time trial was conducted for the T1-class athletes, and a 20 km time trial was conducted for the T2-class athlete. Both sport classes performed the test simultaneously to simulate a competitive race and were encouraged by the coach and researchers.

### 2.9. Training Program

The periodization, dosage, and monitoring of the training load were conducted in collaboration with the athletes’ coach. Additionally, guidance and recommendations on volume, intensity, and exercises applied using the HIIT method were provided by a national coach with international experience (bronze medalist in the Tokyo 2020 Paralympic Games with a T2-class athlete in a road race).

The classic periodization model by Matveev [46,47,48] was used, distributed into three microcycles, with a training frequency of two sessions per week, each lasting 50 min. This study was implemented during the special preparation phase before the 2023 National Para Games in Colombia.

The participants performed all HIIT training sessions using a bicycle ergometer (Avantronic, Cyclus 2^®^, RBM elektronik-automation GmbH, Leipzig, Germany). Based on the aerobic and anaerobic power tests, four power training zones in watts (W) were established to guide the intervention: Zone 1 (40–65%), Zone 2 (70–80%), Zone 3 (85–100%), and Zone 4 (105–120%). Additionally, five heart rate (HR) training zones were defined based on the maximum HR (HR_max_) recorded during the incremental protocol: Zone 1 (50–60%), Zone 2 (60–70%), Zone 3 (70–80%), Zone 4 (80–90%), and Zone 5 (90–100%) [11].

Table 1 presents the workload distribution during the HIIT sessions throughout the intervention. The training program consisted of eight sessions (long HIIT: four sessions; short HIIT: four sessions). The training load was distributed following the principles of individuality. The training intensity was determined using the results from the incremental test to prescribe long HIIT sessions and the WAnT_30 s test to establish short HIIT sessions, both measured in watts (W). Although the intensity was similar for all three athletes, the training volume varied based on their individual test results.

The prescribed volume for long HIIT consisted of 2 sets of 5 repetitions of a 3 min work cycle, with a 3 min recovery between repetitions. During long HIIT sessions, the intensity for the work intervals was set at 85% W (Athlete A: 123.3 W; Athlete B: 123.3 W; Athlete C: 191.3 W), while the recovery intervals were performed at 60% W (Athlete A: 87 W; Athlete B: 87 W; Athlete C: 135 W).

For short HIIT, the programmed volume consisted of 1 set of 10 repetitions of a 30 s work cycle, with a 1 min recovery between repetitions. The intensity for short HIIT work intervals was set at 100% W (Athlete A: 282.3 W; Athlete B: 272.5 W; Athlete C: 473.7 W), while the recovery intervals were performed at 20% W (Athlete A: 56.5 W; Athlete B: 54.5 W; Athlete C: 94.7 W).

During training sessions, heart rate was continuously monitored, and athletes were asked to report their rate of perceived exertion (RPE) using a modified Borg scale ranging from 0 to 10, where 0 indicates minimal intensity and 10 represents maximal intensity. Athletes were familiarized with this scale before this study.

### 2.10. Statistical Analysis

The statistical analysis was performed using SPSS software (Version 24.0, licensed by the Autonomous University Corporation of Cauca). The data are presented individually for three athletes only. Descriptive statistics were applied using measures of central tendency (mean) and the coefficient of variation (*CV* = [Standard Deviation/Mean] × 100). The percentage of change was calculated using the following formula: Δ = [(posttest − pretest)/pretest] × 100 [27].

## 3. Results

The body composition characteristics of the participants are shown in Table 2. However, these variables exhibited minimal percentage changes. Body weight and BMI decreased by 0.2% to 0.6% in Athletes A, B, and C. Muscle mass increased by only 0.8%, while body fat showed a posttest difference of 3.2% in Athlete A.

It is important to highlight that all three athletes were following a structured training program prior to this study, with the training volume monitored through the total weekly kilometers. Subsequently, a systematized training methodology was implemented and adjusted by the coach based on the individual characteristics of each participant.

The estimated VO_2_max and anaerobic power results for both the pretest and posttest are presented in Table 3. In terms of the estimated VO_2_max derived from the incremental test, percentage improvements were observed in Athletes A and C, with increases of 7% and 9.4%, respectively. Athlete B showed no improvement in this variable. Additionally, increases in power output during the incremental test were observed only in Athletes A and C.

Improvements in the maximum power output (maximum watts) during the WAnT_30 s test after the HIIT program were observed in all participants: 31% (282.3 vs. 370.4 W) in Athlete A, 15% (272.5 vs. 312.6 W) in Athlete B, and 9% (473.7 vs. 516.2 W) in Athlete C. The mean power output increased by 43% in Athlete A. The fatigue index also demonstrated favorable changes in all three athletes. Anaerobic power, defined as the ratio between maximum power and body weight, increased by between 5.3% and 30.7% across all participants.

Anaerobic capacity, measured as the ratio of mean power to body weight, increased by 42.5% in Athlete A, 9% in Athlete B, and 1.5% in Athlete C. Positive numerical and percentage changes were observed in all analyzed variables in the posttest.

Table 4 presents the values for the sports performance variable after the HIIT training program. These data are expressed in seconds for the individual time trial races. Overall, a reduction in time was observed in the 14 km time trial for Athlete A (2492 vs. 2325 s, Δ = −6.7%) and Athlete B (2486 vs. 2390 s, Δ = −3.7%). Athlete C showed an improvement in their 20 km time trial performance (2775 vs. 2674 s, Δ = –3.6%, CV: 1.8%).

The physiological response of the cardiovascular system, monitored through heart rate (HR), and the perceptual response, assessed by the rating of perceived exertion (RPE), in paracycling athletes following an eight-session HIIT training program over four weeks are presented below. Table 1 illustrates the behavior of these variables for each athlete.

HR monitoring showed an increase in Athlete A (97–100% HR) from week 1 to week 4, while Athlete B experienced a decrease in HR throughout this study. Athlete C exhibited elevated peaks above 95% HR in weeks 1, 3, and 4.

Similarly, the RPE curve remained stable during the first two weeks but increased in the third and fourth weeks due to the higher intensity of the short HIIT sessions in all participants. Overall, HR fluctuated above 94%, and the RPE remained above 7/10 throughout this study.

## 4. Discussion

The purpose of this research was to analyze the effects of high-intensity interval training (HIIT) on the estimated VO_2_max, anaerobic power, and athletic performance in cyclists with cerebral palsy after a four-week training program. Percentage improvements were observed in the estimated VO_2_max, peak power output (from both the WAnT_30 s and the incremental test), and time trial performance. These enhancements were reflected in reduced completion times for the 14 km individual time trial for Athletes A and B and for the 20 km time trial for Athlete C. All athletes completed both the pre- and postintervention assessments and fully attended the scheduled training sessions.

To our knowledge, this is the first study conducted with paracycling athletes in the T1 and T2 sport classes [23]. The results of the current study show that active participation in paracycling training programs is predominantly male, findings consistent with those reported by Kim et al. [20], who included a similar sample.

The body weight of the participants was lower compared to in the study by Nevin et al. [18] which included physically disabled athletes (69.4 ± 15.4 kg) who underwent an 8-week handcycling training program. The age, weight, and height of the participants in the present study differ from those reported in previous research [49]. The BMI at the end of the HIIT training showed no changes, consistent with the findings of Burkett and Mellifint, who studied Paralympic cyclists using a six-week cycloergometer training program. In their study, the workload was concentrated in 2 min maximal efforts, with partial recovery between intervals set at 65% (114–134 bpm) of heart rate and a total recovery time of 11 min.

The muscle tissue observed in this study was higher compared to previous research [14]. Anaerobic power improved after the intervention, with Athletes A and B showing lower *W* values, while Athlete C exhibited higher values. These results align with the reduced recruitment of muscle fibers for contraction and the limited contribution of phosphagen and glycolytic metabolism in Athletes A and B due to the severity of their disability compared to Athlete C.

Athlete B did not show improvement in the estimated VO_2_max, in contrast to Athletes A and C. This is explained by the adopted motor control function, peripheral factors, poor muscle development, coordination difficulties, and, to a lesser extent, the cardiorespiratory system, which can limit aerobic capacity in athletes with cerebral palsy [50,51]. This outcome may be due to the lack of sufficient implementation of long-duration HIIT intervals, which are critical for enhancing fatigue resistance during prolonged efforts. This is particularly relevant when considering training volume, intensity, and optimal recovery. As illustrated in Table 1, when comparing Athletes A and B (both in the T1 classification), Athlete B exhibited a lower average power output during the long HIIT sessions in the first two weeks. However, in the WAnT_30 s test, Athlete B demonstrated a stronger response, suggesting a greater affinity for efforts relying on glycolytic and phosphagen energy systems, as evidenced by a 15% improvement compared to the baseline test.

Athletes A and C showed improvements in both the estimated VO_2_max and WAnT_30 s, which could be attributed to their greater muscle mass, likely contributing to enhanced motor unit recruitment. Furthermore, the training program may have better matched their individual physiological needs and characteristics. Overall, as a result of the athletes’ adaptation (in some cases) to the training stimulus, reflected in both the estimated VO_2_max and WAnT_30s data, reductions in time trial completion times were observed for the prescribed distances.

The findings of this research show improvements in anaerobic power, particularly in maximum watts (*W*_max_) in the WAnT_30s after the HIIT program, which increased by 31% in Athlete A (282.3 vs. 370.4 W), 15% in Athlete B (272.5 vs. 312.6 W), and 9% in Athlete C (473.7 vs. 516.2 W) compared to the pretest. Additionally, the mean power improved by 43% in Athlete A (218.4 vs. 312.2 W), 9% in Athlete B (226.6 vs. 246.6 W), and 16% in Athlete C (388.3 vs. 451.3 W). These findings indicate that the HIIT training program was effective in enhancing anaerobic power in all athletes.

Anaerobic power in the study by Furno Puglia et al. [45], involving international handcycling athletes, ranged from 223.9 to 571.7 W, aligning with the maximum watts (*W*_max_) values reported in this study (277.4 to 516.2 W). However, these results are also comparable to those found in football players with cerebral palsy, who achieved a maximum power output of 490.6 ± 125.7 W [52]. It is noteworthy that these data surpass those of Athletes A and B, but the maximum watts (*W*_max_) generated by Athlete C were higher (516.2 W). In contrast, trained cyclists completed the same test with a power output of 799.2 ± 267.0 W [53].

In the study by Yanci et al. [52], the mean power, fatigue index, and anaerobic power were superior, as evidenced by higher maximum watts (*W*_max_) values mobilized within the first 6 s of the test, where the mean *W* mobilized remained close to the maximum watts (*W*_max_) achieved. It is important to pay attention to the fatigue index, since it is a variable that allows for the control of the athletes so that the maximum power levels do not fall considerably during the WAnT_30 s.

In this context, better performance was observed in the WAnT_30 s, which is of short duration, compared to the incremental protocol, which is a prolonged test. These positive numerical and percentage changes in aerobic endurance and maximum watts (*W*_max_) can be explained by the fact that the athletes followed both long HIIT and short HIIT training programs on the bicycle ergometer during this study.

Similar results were reported by Koontz et al. [29] and Russo et al. [54], where percentage and numerical changes in the VO_2_max were observed after a HIIT program in handcycling.

Additionally, Russo et al. [54], in an incremental test with five handcycling athletes, found maximum watts (*W*_max_) values similar to those mobilized by the athletes in the current study but lower than those observed in non-disabled cyclists who participated in the same experiment. Previous studies [10,17] have reported that the heart rate (HR) response was higher during *W* production in incremental tests, with positive changes observed in all athletes.

Regarding sports performance variables, there was a reduction in the time required to complete the individual time trials of 14 km and 20 km for Athletes A and B and Athlete C, respectively. Athletes A and B improved their individual performance in the 14 km time trial, a finding that aligns with those reported by Kim et al. [20], who applied the same test to physically disabled athletes at an altitude of 4000 m above sea level.

In contrast to the present study, Boer and Terblanche [55] and Flueck [14] reported lower completion times (2220 s) for a 20 km time trial in T2-class athletes. These differences could be attributed to factors such as variations in sample age and body composition, with the athletes in their study presenting a lower proportion of fat tissue and a higher proportion of lean muscle tissue. Additionally, these factors influence workload dosing and training program design, which may explain the discrepancies in the results.

The data collected on the physiological response through heart rate (HR) and the subjective response via the rate of perceived exertion (RPE) during the eight training sessions enabled effective workload control and optimization. This approach helped monitor and mitigate states of overload and overtraining, reducing the risk of potential injuries in paracycling athletes [35,56]. Table 1 illustrates the HR and RPE trends during the intervention, showing that athletes did not experience adverse effects by the end of this study. They were able to sustain a high workload mobilization while maintaining an optimal physiological response, which suggests their capacity to withstand prolonged efforts and delay the onset of fatigue [56].

However, this study presents certain limitations. The small sample size, the absence of female athletes, and the lack of control groups represent significant constraints. Given the heterogeneity among athletes, sport classes, and types of disabilities, the findings may vary. These challenges stem from the difficulty of accessing this population due to territorial dispersion and their specific characteristics. Future studies should consider working with athletes residing in institutional settings to better control the variables under investigation.

Another limitation of this study is that, as a non-controlled quasi-experimental design, biochemical and physiological markers were not directly measured [57]. Future research should incorporate these techniques to enhance the accuracy of physiological assessments. Moreover, training programs should prioritize and integrate strength, flexibility, and road-based training components to improve the transfer and assimilation of training loads applied in laboratory settings. Additionally, the equation used to estimate the VO_2_max has not been previously validated in athletes with cerebral palsy; therefore, the findings should be interpreted with caution.

No adverse effects were reported during or after this study, as internal training load monitoring was conducted using technological devices. These tools allowed for the real-time tracking of power output and cardiovascular response, helping to reduce fatigue and prevent any worsening of the athletes’ disabilities [56]. According to the American College of Sports Medicine [58], interventions for this population should focus on short training sessions, emphasizing closed kinetic chain resistance exercises to enhance muscle strength and prevent epileptic episodes and seizures [59,60].

Longer intervention studies are necessary to confirm these findings over the medium and long term. In this study, one of the main limitations preventing an extension beyond four weeks was the cyclists’ national competition schedule. Despite this constraint, this study stands out for its novelty, as it paves the way for more exhaustive, controlled, and rigorous future research.

Additionally, while this is the first study conducted with T1- and T2-class athletes, future research should incorporate additional measurement protocols to thoroughly analyze the effects on body composition, physical fitness, and sports performance over longer periods. This will contribute to identifying the optimal training approach for paracycling athletes.

## 5. Conclusions

In conclusion, this study found that high-intensity interval training (HIIT) had a positive impact on the estimated VO_2_max in Athletes A and C. Additionally, improvements in anaerobic power output and athletic performance were observed in all three athletes over a short training period of four weeks. These gains were reflected in reduced time trial durations in both the 14 km and 20 km events, specific to each athlete’s sport class.

## Figures and Tables

**Table 1 jfmk-10-00102-t001:** Training program during the four weeks of HIIT and HR monitoring and RPE response over four weeks of HIIT in paracycling athletes.

Week—HIIT Type	Sessions	Athlete A				Athlete B				Athlete C			
		Planned Exercise Intensity (W)	Performed Mean Exercise Intensity (W)	HR (%)	RPE	Planned Exercise Intensity (W)	Performed Mean Exercise Intensity (W)	HR (%)	RPE	Planned Exercise Intensity (W)	Performed Mean Exercise Intensity (W)	HR (%)	RPE
1—Long	1	123.3 (85)	86 (59)	170 (95)	7	123.3 (85)	86 (70)	175 (93)	8	191.3 (85)	146 (65)	170 (99)	7
	2	123.3 (85)	97 (67)	175 (98)	7	123.3 (85)	97 (79)	186 (98)	6	191.3 (85)	149 (66)	161 (94)	7
2—Long	3	123.3 (85)	109 (75)	174 (97)	8	123.3 (85)	86 (70)	177 (94)	8	191.3 (85)	153 (68)	160 (93)	8
	4	123.3 (85)	112 (77)	176 (99)	6	123.3 (85)	98 (80)	180 (95)	8	191.3 (85)	150 (67)	166 (97)	8
3—Short	5	282.3 (100)	73 (26)	179 (100)	10	272.5 (100)	71 (26)	181 (96)	8	473.7 (100)	112 (24)	164 (95)	8
	6	282.3 (100)	83 (29)	176 (99)	8	272.5 (100)	72 (26)	175 (93)	6	473.7 (100)	134 (28)	166 (97)	8
4—Short	7	282.3 (100)	113 (40)	177 (99)	9	272.5 (100)	101 (37)	174 (92)	9	473.7 (100)	147 (31)	168 (98)	9
	8	282.3 (100)	130 (46)	179 (100)	9	272.5 (100)	112 (41)	178 (95)	7	473.7 (100)	150 (32)	170 (99)	9

Planned exercise intensity: watts programmed; performed mean exercise intensity: average watts during each training session; W: watts of power; %: percentage; HR: heart rate. The long HIIT sessions were programmed based on the results of the incremental test, while the short HIIT sessions were programmed using data from the WAnT_30 s test.

**Table 2 jfmk-10-00102-t002:** Summary of body composition data before and after the HIIT program.

Characteristic Protocol	Athlete A				Athlete B				Athlete C			
	*Pre*	*Post*	Δ	*CV*	*Pre*	*Post*	Δ	*CV*	*Pre*	*Post*	Δ	*CV*
Body Composition												
Body weight (kg)	54.8	54.5	0.6	0.31	51.4	51.1	0.5	0	72.4	72.1	0.3	0.21
BMI (kg/m^2^)	22.5	22.3	0.6	0.31	19.8	19.7	0.6	0	24.7	24.6	0.4	0.2
Muscle mass (%)	48.6	49.0	0.8	0.41	43.9	44.2	0.7	0	59.7	59.9	0.3	0.17
Body fat (%)	9.3	9.0	3.2	1.64	10.2	10.0	1.9	0	11.1	11	0.9	0.45

*Pre*: pretest; *Post*: posttest; Δ: percentage change; *CV*: coefficient of variation; kilograms; BMI: Body Mass Index; %: percentage.

**Table 3 jfmk-10-00102-t003:** Summary of estimated VO_2_max and anaerobic power data before and after the HIIT program.

Characteristic Protocol	Athlete A				Athlete B				Athlete C			
	*Pre*	*Post*	Δ	*CV*	*Pre*	*Post*	Δ	*CV*	*Pre*	*Post*	Δ	*CV*
**Estimated VO_2_max**												
VO_2_max (ml·kg^−1^ min^−1^)	2192	2347	7.0	3.4	2128	2128	0.0	0.0	2756	3018	9.4	4.6
Max power (*W*_max_)	145	160	10.3	4.9	145	145	0.0	0.0	225	250	11.1	5.3
HR_max_ (bpm)	178	183	2.8	1.4	188	185	1.6	0.8	171	173	1.1	0.6
**WAnT_30 s**												
Max power peak (*W*_max_)	282.3	370.4	31.2	13.5	272.5	312.6	15	6.9	473.7	516.2	9.0	4.4
Mean power (W)	218.4	312.2	43	17.7	226.6	246.6	9.0	4.2	388.3	451.3	16.2	7.5
Fatigue index (W/s)	2.0	5.2	160	44.4	8.3	9	8.4	4.0	65.0	7.1	65.2	80.7
Anaerobic power (W/kg)	5.2	6.8	30.7	13.3	5.3	6.1	15	7.0	6.7	7.1	5.9	2.9
Anaerobic capacity (W/kg)	4.0	5.7	42.5	17.5	4.4	4.8	9.0	4.4	6.3	6.2	1.5	0.8

*Pre*: pretest; *Post*: posttest; Δ: percentage change; *CV*: coefficient of variation (%); W: watts of power; bpm: beats per minute.

**Table 4 jfmk-10-00102-t004:** Summary of sports performance data before and after the HIIT program.

Athlete A 14 km Time Trial (sec)				Athlete B 14 km Time Trial (sec)				Athlete C 14 km Time Trial (sec)			
*Pre*	*Post*	Δ	*CV* (%)	*Pre*	*Post*	Δ	*CV* (%)	*Pre*	*Post*	Δ	*CV* (%)
2492	2325	−6.7	3.4	2486	2390	−3.7	1.9	2775	2674	−3.6	1.8

*Pre*: pretest; *Post*: posttest; Δ: percentage change; *CV*: coefficient of variation (%); km: kilometers; sec: seconds.

## Data Availability

The data presented in this study are available upon request from the corresponding author.

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
