# Peer review of "Effect of 4 Weeks of High-Intensity Interval Training (HIIT) on VO2max, Anaerobic Power, and Specific Performance in Cyclists with Cerebral Palsy"

_jfmk, 2025, doi:10.3390/jfmk10020102_

Round 1
Reviewer 1 Report
Comments and Suggestions for Authors
There was no mention of adverse effects and future studies that talk about the occurrence or not of adverse effects and the cardioprotective effect such as the use of HRV available in the equipment used in the H10 polar collection or the effect on the HIIT follow-up.

Author Response
Para artículo de investigación
Respuesta al revisor 1 Comentarios
|
||
1. Resumen |
|
|
Muchas gracias por tomarse el tiempo de revisar este manuscrito. A continuación, encontrará las respuestas detalladas y las revisiones o correcciones correspondientes resaltadas o con control de cambios en los archivos reenviados .
|
||
2. Preguntas para la evaluación general |
Evaluación del revisor |
Respuesta y revisiones |
¿La introducción proporciona suficiente información de fondo e incluye todas las referencias pertinentes? |
Sí/Se puede mejorar/Se debe mejorar/No aplica |
[Si es necesario, por favor, responda. O también puede dar su respuesta correspondiente en la carta de respuesta punto por punto. La misma que se muestra a continuación] |
¿Todas las referencias citadas son relevantes para la investigación? |
Sí/Se puede mejorar/Se debe mejorar/No aplica |
|
¿Es apropiado el diseño de la investigación? |
Sí/Se puede mejorar/Se debe mejorar/No aplica |
|
¿Están adecuadamente descritos los métodos? |
Sí/Se puede mejorar/Se debe mejorar/No aplica |
|
¿Están claramente presentados los resultados? |
Sí/Se puede mejorar/Se debe mejorar/No aplica |
|
¿Las conclusiones están respaldadas por los resultados? |
Sí/Se puede mejorar/Se debe mejorar/No aplica |
|
3. Respuesta punto por punto a los Comentarios y Sugerencias para Autores |
||
Comentarios 1: Materiales y Métodos, agregar que los participantes pueden abandonar el estudio sin ninguna pérdida, Línea 119-120. |
||
Respuesta 1 : Gracias por señalarlo. Estoy/estamos de acuerdo con este comentario. Se les informó del propósito de la investigación, se familiarizaron con los procedimientos, riesgos y beneficios y podían retirarse de la investigación en cualquier momento sin ninguna pérdida.
|
||
Comentarios 2: Resultados, Eliminar, Línea 222-224. |
||
Respuesta 2: Gracias por señalarlo. Estoy/estamos de acuerdo con este comentario. Por lo tanto, aceptamos la eliminación de las líneas 222 a 224 que contienen la siguiente información: Esta sección puede dividirse en subtítulos. Debe proporcionar una descripción concisa y precisa de los resultados experimentales, su interpretación, así como las conclusiones experimentales que se pueden extraer. |
||
Comentarios 3: Resultados, Sería interesante tener datos de entrenamiento estándar de estos atletas para comparar, Línea 227-228. |
||
Respuesta 3: Gracias por señalarlo. Estoy/estamos de acuerdo con este comentario. El peso corporal y el IMC disminuyeron en un 0,3%, la masa muscular aumentó solo un 0,2% y la grasa corporal mostró una diferencia de 1,3% en el post-test. La altura no presentó cambios. Es importante destacar que los tres atletas siguieron antes del estudio un programa de entrenamiento dosificado en volumen de entrenamiento con la suma de kilómetros semanales, la intensidad de la carga de trabajo fue monitoreada por sensaciones, luego de lo cual se implementa una metodología de trabajo sistematizada y ajustada por el entrenador dadas las características individuales de cada participante. |
||
Comentarios 4: Resultados, ¿Es necesario mantener? la intervención no tiene impacto en esta variable, Tabla 2. |
||
Respuesta 4: Gracias por señalarlo. Estoy/estamos de acuerdo con este comentario. Esta información se elimina de la tabla 2. |
||
Comentarios 5: [Discusión, Describir el período de intervención y la modalidad HIIT, Línea 291-292] |
||
Respuesta 5: Gracias por señalar esto. [ ]” |
||
Comentarios 6: Discusión, describa el periodo de intervención y modalidad HIIT, Línea 320-322. |
||
Respuesta 6: Gracias por señalarlo. Estoy/estamos de acuerdo con este comentario. La información solicitada se complementa con: El IMC al final del entrenamiento HIIT no mostró cambios, según lo reportado por Burkett y Mellifint 42 en atletas de ciclismo paralímpico utilizando seis semanas de entrenamiento en cicloergómetro, la carga de trabajo se concentró en esfuerzos máximos de 2 minutos, la recuperación parcial entre intervalos se estableció en 65% (114 - 134 lpm) de la frecuencia cardíaca, la recuperación total fue de 11 minutos. |
||
Comentarios 7: [Discusión, Metodología de intervención muy diferente a la propuesta, sugiero eliminar, Línea 320-322). |
||
Respuesta 7: Gracias por señalar esto. Según nuestros hallazgos, Kim et al. [12] observaron una disminución del VO2máx en 2 de 320 3 atletas estudiados durante el mismo período mientras utilizaban entrenamiento de hipoxia hipobárica a una altitud de 4000 m. |
||
Comentarios 7: Discusión, discutir si ocurrieron efectos adversos en el estudio, Línea 344-345. |
||
Respuesta 7: Gracias por señalarlo. Estoy/estamos de acuerdo con este comentario. La Figura 1 muestra una disminución en la curva de FC y RPE durante la intervención, lo que sugiere que los atletas no presentaron efectos adversos al final del estudio, ya que fueron capaces de realizar una alta movilización de la carga de trabajo, manteniendo una respuesta fisiológica óptima para soportar esfuerzos prolongados desafiando la aparición de la fatiga. |
||
Comentarios 8: Discusión, describa más sobre la ocurrencia o no de efectos adversos y el efecto cardioprotector, como el uso de HRV disponible en el equipo utilizado para recolectar el H10 polar, o el efecto en el seguimiento del HIIT, Línea 357. No se hizo mención de efectos adversos y estudios futuros que hablen de la ocurrencia o no de efectos adversos y el efecto cardioprotector como el uso de HRV disponible en los equipos utilizados en la colección polar H10 o el efecto en el seguimiento del HIIT. |
||
Respuesta 8: Gracias por señalar esto. No se reportaron efectos adversos durante y después del estudio, debido a que el monitoreo de la carga interna de entrenamiento se realizó con dispositivos tecnológicos que contribuyen a generar un monitoreo en tiempo real de la potencia desarrollada y la respuesta cardiovascular, esto permite reducir el estado de fatiga y no agravar el grado de discapacidad en deportistas con parálisis cerebral. Según el Colegio Americano de Medicina del Deporte, las intervenciones con esta población deben estar dirigidas a sesiones cortas de trabajo, donde se priorice el entrenamiento de resistencia en cadena cinética cerrada debido a que hay mejoría en la fuerza muscular y previene episodios de epilepsia y convulsiones. |
||
Comentarios 9: Conclusiones, Destacar los resultados, como la potencia media en 16,2%, el índice de fatiga en 30,4%, la potencia anaeróbica en 16,2% y la capacidad anaeróbica en 13,4%. Se observa reducción de tiempo para la contrarreloj de 14 km (Δ = 6,05%) y la contrarreloj de 20 km (Δ = 3,6%), Recta 364-366. |
||
Respuesta 9: Gracias por señalar esto. En conclusión, este estudio informó que el HIIT influyó positivamente en el VO2max, mejoras moderadas en la producción de potencia anaeróbica y el rendimiento atlético en atletas de clase T1-T2 durante un corto período de cuatro semanas. Estos cambios se reflejaron en duraciones de contrarreloj reducidas para carreras de 14 km y 20 km para cada clase deportiva. Esto se debe a que en la sección de resultados se realiza un resumen de la información. |
||
Comentarios 9: Conclusiones, añadir al final de la discusión, Línea 368-371. |
||
Respuesta 9: Gracias por señalar esto. Si bien este es el primer estudio realizado con atletas de las clases T1 y T2, se recomienda que las investigaciones futuras incluyan protocolos de medición adicionales para analizar en profundidad los efectos sobre la composición corporal, la aptitud física y el rendimiento deportivo a mediano y largo plazo. Esto ayudará a identificar el enfoque de entrenamiento óptimo para los atletas de Paracycling. |
||
|

Reviewer 2 Report
Comments and Suggestions for Authors
Dear authors, my contributions are attached. Thank you for the opportunity to read and comment on the manuscript.
Academic Congratulations

The language needs to be revised.
Author Response
Para artículo de revisión
Respuesta al revisor 2 comentarios
|
||
1. Resumen |
|
|
Muchas gracias por tomarse el tiempo de revisar este manuscrito. A continuación, encontrará las respuestas detalladas y las revisiones o correcciones correspondientes resaltadas o con control de cambios en los archivos reenviados .
|
||
2. Preguntas para la evaluación general |
Evaluación del revisor |
Respuesta y revisiones |
¿Es el trabajo una contribución significativa al campo? |
|
[Si es necesario, por favor, responda. O también puede dar su respuesta correspondiente en la carta de respuesta punto por punto. La misma que se muestra a continuación] |
¿Está el trabajo bien organizado y descrito exhaustivamente? |
|
|
¿El trabajo es científicamente sólido y no engañoso? |
|
|
¿Existen referencias apropiadas y adecuadas a trabajos relacionados y anteriores? |
|
|
¿El inglés utilizado es correcto y legible? |
|
|
3. Respuesta punto por punto a los Comentarios y Sugerencias para Autores |
|
|
Comentarios 1: Resumen, Dar una breve descripción del tema. |
||
Respuesta 1 : Gracias por señalar esto. El HIIT es útil y eficaz para entrenar a atletas paralímpicos con discapacidad cerebral en deportes intermitentes y deportes donde la capacidad aeróbica y anaeróbica, la velocidad y la fuerza son esenciales para retrasar la aparición de la fatiga y optimizar el rendimiento deportivo . |
||
Comentarios 2: Resumen, Materiales y Métodos: describa el diseño del estudio; mencione los términos y procedimientos y resuma lo descrito en la sección de métodos del artículo; describa lo más fielmente posible. |
||
Respuesta 2: Gracias por señalar esto. En este estudio cuasi -experimental participaron tres atletas masculinos (Atleta A, B y C) con parálisis cerebral de la Comisión Departamental de Paraciclismo del Cauca (Colombia). El VO 2máx estimado se determinó mediante una prueba incremental en un cicloergómetro. La potencia anaeróbica se midió utilizando el Test Anaeróbico de Wingate de 30 s de duración (WAnT_30 sec ) . El rendimiento específico se evaluó con una contrarreloj individual de 14 km para la clase T1 y 20 km para la clase T2. El entrenamiento HIIT se realizó en un cicloergómetro durante cuatro semanas (dos sesiones por semana). La intensidad del entrenamiento se basó en Watts (W) medidos en la prueba incremental para HIIT largo y la prueba WAnT_30 sec para HIIT corto. La carga de entrenamiento se monitoreó a través de las respuestas de la frecuencia cardíaca (FC) y el esfuerzo percibido subjetivo (RPE) al final del entrenamiento. |
||
Comentarios 3: Resumen, Conclusión: “Este estudio informó que el HIIT36 influyó positivamente en el VO2máx, mejoras moderadas en la producción de potencia anaeróbica37 y el rendimiento atlético en atletas de clase T1-T2 durante un corto período de cuatro semanas”; sea más enfático, ¿qué mejoró? ¿Velocidad? ¿Poder? ¿Resistencia? |
||
Respuesta 3: Gracias por señalar esto. Este estudio informó que las ocho sesiones de HIIT influyeron positivamente en el VO 2 máximo estimado, mejoras moderadas en la potencia de salida máxima y promedio en WAnT_30 segundos y en el rendimiento en pruebas contrarreloj en todos los atletas durante un corto período de cuatro semanas. |
||
Comentarios 4: Palabras clave, Evite repetir las palabras y términos mencionados en el título. |
||
Respuesta 4: Gracias por señalar esto . Para-deporte, deporte paralímpico, HIIT, atletas ciclistas , rendimiento deportivo. |
||
Comentarios 5: Introducción, se sugiere fuertemente conceptualizar la condición de parálisis cerebral y describir sus limitaciones. |
||
Respuesta 5: Gracias por señalar esto. La parálisis cerebral no es una enfermedad en sí misma, sino un término que engloba una variedad de trastornos motores de origen diverso, que pueden cambiar con la edad4 . Se define como un conjunto de alteraciones permanentes, pero no inmutables, del movimiento y de la función motora, resultantes de una lesión, anomalía o alteración no progresiva en el cerebro en desarrollo. Estos trastornos se manifiestan desde la infancia y persisten durante toda la vida, afectando la postura y el control motor. Aunque la parálisis cerebral no empeora con el tiempo, sus manifestaciones pueden evolucionar con el crecimiento5 . Además de las dificultades motoras, son frecuentes las alteraciones sensoriales, perceptivas y cognitivas, así como la epilepsia y los problemas musculoesqueléticos asociados6 . |
||
Comentarios 6: Introducción, Línea 75: explique en detalle por qué el entrenamiento en intervalos de alta intensidad (HIIT) ha demostrado ser eficaz. Comente algunos de los estudios de la introducción y señale las mejoras en la condición física de la población citada. |
||
Respuesta 6: Gracias por señalar esto. Se ha demostrado que uno de los métodos de entrenamiento más eficientes en el tiempo y aplicable a deportistas con discapacidad física, discapacidad visual o parálisis cerebral es el Entrenamiento Interválico de Alta Intensidad (HIIT). Este es el caso de la revisión realizada por Lasso-Quilindo & Chalapud-Narváez 18 , quienes evidenciaron en diferentes estudios que este método de entrenamiento produce efectos positivos en corto tiempo sobre la capacidad aeróbica y anaeróbica, la composición corporal, así como un mayor reclutamiento de fibras musculares, así mismo pudieron establecer que los cambios son altamente significativos cuando se combina con entrenamiento en circuito, entrenamiento de fuerza, pilates y sprint. En este sentido, se puede afirmar que el HIIT es útil y eficiente para entrenar a deportistas paralímpicos con discapacidad física, visual y parálisis cerebral en deportes intermitentes y deportes donde la capacidad aeróbica y anaeróbica, la velocidad y la fuerza son fundamentales para retrasar la aparición de la fatiga y optimizar el rendimiento deportivo. |
||
Comentarios 7: Introducción, Líneas 94 a 97: se sugiere enfáticamente justificar claramente la importancia del estudio, resaltando su relevancia para el campo clínico y el desempeño humano en individuos con limitaciones. |
||
Respuesta 7: Gracias por señalar esto. El estudio del HIIT en Paraciclismo resulta relevante para optimizar el rendimiento deportivo, ya que mejora el VOâ‚‚max, la potencia anaeróbica y la resistencia muscular, aspectos fundamentales en competiciones de ruta y contrarreloj. Además, permite adaptar el entrenamiento a las necesidades de cada categoría funcional (bicicleta, handcycle, tándem y triciclo), favoreciendo respuestas fisiológicas específicas y reduciendo el riesgo de lesiones. Su impacto en la eficiencia energética contribuye a una mejor tolerancia al esfuerzo y retrasa la fatiga. A nivel científico, la escasez de estudios en este ámbito limita la planificación deportiva basada en la evidencia. Por último, su aplicación no solo optimiza el rendimiento, sino que promueve beneficios para la salud, previniendo comorbilidades asociadas a la discapacidad. |
||
Comentarios 8: Materiales y métodos, acerca de: 2.1. Diseño del estudio; colocar el párrafo sobre cuestiones éticas después de la línea 106, (líneas 121 a 125). |
||
Respuesta 8: Gracias por señalar esto. Coloque el párrafo sobre cuestiones éticas después de la línea 106 (líneas 121 a 125). |
||
Comentarios 9: Materiales y Métodos, Punto: 2.2. Sujetos: se sugiere el término “Participantes”. Describir el momento de la lesión (si aplica), las categorías de los participantes y su edad; la tabla 2 es suficiente con las características físicas. |
||
Respuesta 9: Gracias por señalar esto. El estudio involucró a tres atletas paralímpicos masculinos (atletas A, B y C). Atleta A (años de discapacidad: congénita; clase deportiva: T1; edad: 19 años; experiencia competitiva: 5 años), Atleta B (años de discapacidad: congénita; clase deportiva: T1; edad: 24 años; experiencia competitiva: 4,5 años) y Atleta C (años de discapacidad: 9 años; clase deportiva: T2; edad: 56 años; experiencia competitiva: 8 años ). |
||
Comentarios 10: Materiales y Métodos, Insertar el símbolo de marca registrada en el equipo cuando se mencionen las marcas de los equipos utilizados. |
||
Respuesta 10: Gracias por señalar esto. Antes de iniciar los protocolos (pruebas iniciales y finales) y el entrenamiento (durante la intervención) en el cicloergómetro Cyclus 2 (Avantronic, Cyclus 2, RBM elektronik-automation GmbH , Leipzig, Alemania) . |
||
Comentarios 11: Materiales y métodos, recomendamos encarecidamente eliminar la Tabla 1. Dosificación de la carga de entrenamiento externo durante cuatro semanas de HIIT de la sección de Materiales y métodos. Es recomendable insertar la tabla anterior, Tabla 3. Resumen de los datos de VO2max y potencia anaeróbica antes y después del programa HIIT. La segunda opción es crear un dibujo del control de carga, en cuyo caso insertarlo en la sección de Métodos. |
||
Respuesta 11: Gracias por señalar esto. Las tablas y gráficos se ajustan y los resultados se presentan individualmente. |
||
Comentarios 12: Resultados, Se recomienda encarecidamente mejorar la calidad de la Figura 1. |
||
Respuesta 12: Gracias por señalarlo. Estoy/estamos de acuerdo con este comentario. Las tablas y gráficos se ajustan y los resultados se presentan individualmente. |
||
Comentarios 13: Discusión, Sugerimos insertar subtemas para discutir los resultados de cada participante; Participante 1 - (categoría), escribir en cursiva para identificarlo mejor en el texto. |
||
Respuesta 13: Gracias por señalar esto. Analice los cambios realizados y proporcione las explicaciones o aclaraciones necesarias. Mencione exactamente en qué parte del manuscrito revisado se puede encontrar este cambio: número de página, párrafo y línea. |
||
Comentarios 14: Discusión, Línea 299-302 330-333- Sugerimos poner la posible mejora en porcentajes, algunos términos no dan resultados precisos. |
||
Respuesta 14: Gracias por señalar esto. Analice los cambios realizados y proporcione las explicaciones o aclaraciones necesarias. Mencione exactamente en qué parte del manuscrito revisado se puede encontrar este cambio: número de página, párrafo y línea. |
||
Comentarios 15: Conclusión , se sugiere mayor precisión, ser más enfáticos. ¿Qué es lo que realmente ha mejorado? ¿Se puede utilizar el protocolo para las pruebas mencionadas? |
||
Respuesta 15: Gracias por señalar esto. Analice los cambios realizados y proporcione las explicaciones o aclaraciones necesarias. Mencione exactamente en qué parte del manuscrito revisado se puede encontrar este cambio: número de página, párrafo y línea. |
||
Comentarios 16: Conclusión, líneas 367-369. Sugerimos eliminar esto, ya que podría restar valor al trabajo realizado. |
||
Respuesta 16: Gracias por señalar esto. Estoy/estamos de acuerdo con este comentario. Por lo tanto, lo he/hemos hecho. Esta sección se mueve al final de la sección de discusión. |
||
|

Reviewer 3 Report
Comments and Suggestions for Authors
Effect of 4 weeks of high-intensity interval training (HIIT) on VO2max, anaerobic power, and specific performance in 3 cyclists with cerebral palsy
This study aimed to analyse the effects of four weeks of HIIT on VO2max, anaerobic power, and athletic performance in 3 cyclists with cerebral palsy.
Major concerns:
- Data/results presentations. There are only 3 participants and they possessed unique disability; they are deemed as heterogenous. Given this, it would be optimal to present all the results and tests’ data individually and delete the mean data of the 3 subjects
- Rather than just “Wingate test”, the authors should correctly term the test as the 30 s Wingate Anaerobic cycle (WAnT) test. Note that during the test, there are 3 main variables collected. One is Peak Power (which the highest power obtained within the first few second of the test). Second is Mean Power, which is the average power over the 30 s period. Lastly, is the fatigue index which in the percentage decline in power from the peak to the lowest within the 30 s period. In the present study, this reviewer is unclear what ‘anaerobic power’ and ‘anaerobic capacity’ are the authors referring to; are the authors referring to Peak Power and Mean Power, respectively? Please be detail and explicit.
- In the Discussion section, it is not really useful to compare the present data with other or previous studies that have tested on disabled para-athletes because as we know, and you have also mentioned, that each athlete possessed unique disabilities even within the same category or class of disability. For the present study, it would be more useful to explained in detailed the training programme that each athlete underwent – so that coaches and athletes who read this paper might be able to apply the same or modified their training programme to elicit improvements in their own para-athletes.
- For the aerobic (VO2max), anaerobic (Wingate) and exercise performance (the 14 and 20 km TT) tests used in the present study, it would really be useful for the authors to provide the coefficient of variation of the tests’ outcomes and therefore provide the reader with the range of smallest worthwhile change in all of the above the tests’ measure. This will then help the reader to make a decision whether the percentage or degree of change made by the individual in the said test is greater than the “noise” of the test. Has the 3 participants done these tests before – or is this the first time the participants were being tested. I think the 3 participants should be familiarised to all the three tests, epically the time-trial test. Note that the VO2max measure is not taken from respiratory gas measures but predicted from a maximal incremental exercise test – author should report the aerobic data as “predicted or estimated VO2max”.
- Authors should also report the reliability data of the muscle mass and body fat % because these measures were taken via bioelectrical impedance equipment, which is not totally accurate.
- Table 1. What is the definition of “Power Load” and “Total Volume”? How were power load and total volume calculated? Why is total volume is in (W) and I assume its Watts – this does not make sense. It would be ideal if the mean power out achieved by the 3 individuals during each of their training session is displayed/shown for the readers – to show readers what is actually being planned relative to what actually is performed by the athlete. The frequency or the pattern of the Long or Short HIIT was not well described in the table. Authors should highlight for example, for Training Week 1 were 2 Long HIIT sessions, and for Training week 2, it was one Long and one Short HITT session. Additional query, I don’t see any subtle differences in prescribed “intensity of exercise” for the T1 and T2 athletes – why no difference in the HIIT intensity for different athletes.
Minor issues:
- For all figures and tables, please delete data that referred to “All classes n = 3” or mean data of the 2 or 3 individuals. Please show the individual data of each of the 3 athletes. Table 4. Shouldn’t the post-data be in negative. Also, why there are decimal point for the Class T1? Do not average the 2 athletes – always show each participant data.
- Figure 1. It is more useful to report the 3 participants’ individual exercise HR as a relative % of the individual participant’s HRmax rather than absolute exercise HR in beats/min. The same for the discussion section on page 7 Lines 259 – 267.
- How was the exercise HR and RPE tabulated for each week of training? Is it the average of the mean data of the 2 training sessions per week? Authors need to clarify this.
- Authors mentioned “Matveev” but there is no reference of the same author in the reference section.
- The first paragraph of the Results section should be deleted.
- The model, brand and city, country of the Cyclus 2 cycle ergometer were missing; this information should be reported. Is the power output from the ergometer reliable, and validated and the ergometer calibrated.
None.
Author Response
For review article
Response to Reviewer 3 Comments
|
||
1. Summary |
|
|
Thank you very much for taking the time to review this manuscript. Please find the detailed responses below and the corresponding revisions/corrections highlighted/in track changes in the re-submitted files.
|
||
2. Questions for General Evaluation |
Reviewer’s Evaluation |
Response and Revisions |
Is the work a significant contribution to the field? |
|
[Please give your response if necessary. Or you can also give your corresponding response in the point-by-point response letter. The same as below] |
Is the work well organized and comprehensively described? |
|
|
Is the work scientifically sound and not misleading? |
|
|
Are there appropriate and adequate references to related and previous work? |
|
|
Is the English used correct and readable? |
|
|
3. Point-by-point response to Comments and Suggestions for Authors |
|
|
Comments 1: Data/results presentations. There are only 3 participants and they possessed unique disability; they are deemed as heterogenous. Given this, it would be optimal to present all the results and tests’ data individually and delete the mean data of the 3 subjects. |
||
Response 1: Thank you for pointing this out. I/We agree with this comment. Therefore, I/we have. The data are presented individually in Athlete A, B and C. |
||
Comments 2: Rather than just “Wingate test”, the authors should correctly term the test as the 30 s Wingate Anaerobic cycle (WAnT) test. Note that during the test, there are 3 main variables collected. One is Peak Power (which the highest power obtained within the first few second of the test). Second is Mean Power, which is the average power over the 30 s period. Lastly, is the fatigue index which in the percentage decline in power from the peak to the lowest within the 30 s period. In the present study, this reviewer is unclear what ‘anaerobic power’ and ‘anaerobic capacity’ are the authors referring to; are the authors referring to Peak Power and Mean Power, respectively? Please be detail and explicit. |
||
Response 2: Thank you for pointing this out. I/We agree with this comment. Therefore, I/we have. For this study the test is called the 30-s-long Wingate Anaerobic Test (WAnT_30s); The “anaerobic power” refers to the division of the maximum power over the athlete's body weight. Aerobic capacity” refers to the division of the average power over the athlete's body weight. |
||
Comments 3: Discussion, In the Discussion section, it is not really useful to compare the present data with other or previous studies that have tested on disabled para-athletes because as we know, and you have also mentioned, that each athlete possessed unique disabilities even within the same category or class of disability. For the present study, it would be more useful to explained in detailed the training programme that each athlete underwent – so that coaches and athletes who read this paper might be able to apply the same or modified their training programme to elicit improvements in their own para-athletes. |
||
Response 3: Thank you for pointing this out. I/We agree with this comment. Therefore, I/we have. It is important to note that each athlete has unique characteristics. Therefore, a comparison with the results of current studies allows us to expose the scientific evidence on research with similar characteristics to ours. Although the training program is explained in the section on materials and methods, it should be mentioned that the training load for each of the athletes was carried out under the principle of individuality, adapting the tests to the characteristics of the athletes. And from this instance the systematic and individual distribution of the load for the training sessions is made from the incremental test (distribution for long HIIT) and Wingate anaerobic cycle test of 30 s (WAnT) for short HIIT. |
||
Comments 4: For the aerobic (VO2max), anaerobic (Wingate) and exercise performance (the 14 and 20 km TT) tests used in the present study, it would really be useful for the authors to provide the coefficient of variation of the tests’ outcomes and therefore provide the reader with the range of smallest worthwhile change in all of the above the tests’ measure. This will then help the reader to make a decision whether the percentage or degree of change made by the individual in the said test is greater than the “noise” of the test. Has the 3 participants done these tests before – or is this the first time the participants were being tested. I think the 3 participants should be familiarised to all the three tests, epically the time-trial test. Note that the VO2max measure is not taken from respiratory gas measures but predicted from a maximal incremental exercise test – author should report the aerobic data as “predicted or estimated VO2max”. |
||
Response 4: Thank you for pointing this out. I/We agree with this comment. Therefore, I/we have. The tables in the results section show the coefficient of variation.; Previously, the athletes were familiarized with laboratory measurement tests. The time trial races are part of their regular training, and this range of distances are the ones established for national and international competitions.; The change is made for the “estimated VO2max”. |
||
Comments 5: Authors should also report the reliability data of the muscle mass and body fat % because these measures were taken via bioelectrical impedance equipment, which is not totally accurate. |
||
Response 5: Thank you for pointing this out. I/We agree with this comment. Therefore, I/we have. Body composition was assessed using bioelectrical impedance (OMRON Healthcare Technology Co, Ltd., HBF-514C® scale, Kyoto, Japan). The device sends a light electrical current (50 kHz) through the body via electrodes. It is important to mention that this device has been used in previous studies as noted in the section on measurement of body composition. |
||
Comments 6: Table 1. What is the definition of “Power Load” and “Total Volume”? How were power load and total volume calculated? Why is total volume is in (W) and I assume its Watts – this does not make sense. It would be ideal if the mean power out achieved by the 3 individuals during each of their training session is displayed/shown for the readers – to show readers what is actually being planned relative to what actually is performed by the athlete. The frequency or the pattern of the Long or Short HIIT was not well described in the table. Authors should highlight for example, for Training Week 1 were 2 Long HIIT sessions, and for Training week 2, it was one Long and one Short HITT session. Additional query, I don’t see any subtle differences in prescribed “intensity of exercise” for the T1 and T2 athletes – why no difference in the HIIT intensity for different athletes. |
||
Response 6: Thank you for pointing this out. I/We agree with this comment. Therefore, I/we have. According to the request, table 1 is adjusted. It presents a better description of the workload implemented individually week by week, provides information on the types of HIIT and the average watts mobilized during the training sessions. It is essential to note that the intensity is similar for each athlete, but the volume of work mobilized in Watts changes, respecting the principle of individuality. |
||
Comments 7: For all figures and tables, please delete data that referred to “All classes n = 3” or mean data of the 2 or 3 individuals. Please show the individual data of each of the 3 athletes. Table 4. Shouldn’t the post-data be in negative. Also, why there are decimal point for the Class T1? Do not average the 2 athletes – always show each participant data. |
||
Response 7: Thank you for pointing this out. I/We agree with this comment. Therefore, I/we have. In each table and for the analysis, the individual results are presented. For table 4, the values are indeed negative. |
||
Comments 8: Figure 1. It is more useful to report the 3 participants’ individual exercise HR as a relative % of the individual participant’s HRmax rather than absolute exercise HR in beats/min. The same for the discussion section on page 7 Lines 259 – 267. |
||
Response 8: Thank you for pointing this out. Figure 1 shows 2 graphs showing HR and RPE individually for each athlete. |
||
Comments 9: How was the exercise HR and RPE tabulated for each week of training? Is it the average of the mean data of the 2 training sessions per week? Authors need to clarify this. |
||
Response 9: Thank you for pointing this out. Figure 1 shows two graphs where the HR and RPE are plotted individually for each athlete averaged week by week. These results are percentage values of the maximum HR obtained in the incremental test. |
||
Comments 10: Authors mentioned “Matveev” but there is no reference of the same author in the reference section. |
||
Response 10: Thank you for pointing this out. I/We agree with this comment. Therefore, I/we have. Due citation is made to the main author who pointed out the periodization model. |
||
Comments 11: The first paragraph of the Results section should be deleted. |
||
Response 11: Thank you for pointing this out. I/We agree with this comment. Therefore, I/we have. The suggested section is eliminated. |
||
Comments 12: The model, brand and city, country of the Cyclus 2 cycle ergometer were missing; this information should be reported. Is the power output from the ergometer reliable, and validated and the ergometer calibrated. |
||
Response 12: Thank you for pointing this out. I/We agree with this comment. Therefore, I/we have. The requested device information is integrated. |
||
|

Round 2
Reviewer 3 Report
Comments and Suggestions for Authors
- Throughout the manuscript. Please use decimal point (.) and not comma (,) for numeral values. Also, standardized the number of decimal points. For example in Table 3, sometimes you used 2 and sometimes 3 and sometimes no decimal point – please be consistent.
- Line 179-180. The authors replied to my initial comments: “Body composition was assessed using bioelectrical impedance (OMRON Healthcare Technology Co, Ltd., HBF-514C® scale, Kyoto, Japan). It is important to mention that this device has been used in previous studies as noted in the section on measurement of body composition.” The exact information that this reviewer is seeking is whether the equipment has previously been validated and the level of reliability of the equipment.
- Line 197. The regression equation was obtained from reference no 37. How was this regression equation obtained, i.e., from abled or disabled sample of participants? If disabled, is it from participants who are cerebral palsy. Basically, reviewer is seeking confirmation that the regression equation to determine VO2max is valid for use for the 3 participants of this study.
- Line 232. What type/model of cycle ergometer was sued during the 3 participants’ training session.
- Line 239 and 240. “The training load ….. 3 minutes at 85% W, with a 3 min recovery …..60% W. Short HIIT at 100% W, with a …..at 20% W”. Authors needs to be very clear with their writing; for in the above sentence, the values of at 85% W, 60% W, 100% W and 20% W of what reference values. As it currently stand, the sentence is unclear.
- Table 1 is unclear. What is “Programmed” and what is “Media? Also, the values of the power output should be shown in both Watts as well as a % of relative. Line 268 to 172. This entire paragraph caused some confusion. Firstly, author stated that the 3 athletes followed a “structured training program”, and then said that the intensity was monitored through subjective perception (but this reviewer has the impression that training intensity was based on relative percentage of the athlete’s performances in the aerobic and anaerobic test, with values shown in Table 1). Then the authors mentioned that “a systematized training methodology was implemented and adjusted by the coach according to the individual characteristics of each participant”, again the reviewer have the impression that training load/intensity was already fixed according to the relative abilities of the participants (reviewer is referring to Table 1 again). Again, as I previous requested that the mean power output in the Wingate test achieved by the 3 individuals during each of their training session is displayed/shown to the readers – to provide readers as to what is actually being planned relative to what actually is performed by the said athlete. Further, the authors mentioned in their reply to my initial review: “It is essential to note that the intensity is similar for each athlete, but the volume of work mobilized in Watts changes, respecting the principle of individuality.” This is again confusing because the intensity is relative to each individual, but the amount or volume of work, to me at least is fixed – as stated by the from Line 238-242 where log HIIT consisted of “2 sets of 5 reps of 3 min with a 3 min rest”, and for short HIIT is “1 set of 10 reps of 30 s cycle with a 1 min rest between reps”.
- Line 253. Reviewer is concern about the stats in the paper. Firstly, when percentage changes is used in the calculation between pre and post-test for each test for each participant – why author need to include the paired samples significance test? Once you have the percentage changes – you compare the % change with the CV of the test. If the % change in the pre to post test is greater than 2 x CV, then it is deemed that a true or real change has occurred in that individual. If the % change in the pre to post test is less than 2 x CV, then this may or may not signify a true change, the value of the change is within the variability or ‘noise’ of the test. For example, in the changes in the estimated VO2max measure, athlete B clearly did not improve. And for the time-trial, Athlete B and C improves by a margin of 3.7%; is this small margin of improvements a true and real indication of actual performance improvements or again its merely within the variability or ‘noise’ of the time-trial test.
- Line 254. You should have only one CV value for each test. How was CV of each of the test calculated?
- Figure 1 A and B is not useful at all. The 2 figures merely show the comparisons of the 3 athletes’ training HR and RPE, but alas, given the differences between the 3 athletes (as we have agreed) – the figures’ data are meaningless. It would have been better to show the HR, RPE and Power Output of each of the training session for each individual athletes across the 8 training sessions rather than averaging the session over a week.
- Table 2 and Table 3. I don’t understand what are the effects size (ES) comparing between which measure. To my knowledge, ES is used to compare between 2 measures. Inconsistency in Table 3 with sometimes the (,) and (.), please be consistent.
- Write max HR as HRmax.
- The discussion section should primarily be about whether each athlete has performed the training that has been tasked upon him/her and whether there is actually ‘real’ improvements made by the athlete (this should be based on the percentage changes of performance relative to the CV of the test). For example, it will be appropriate and relevant to discuss why Athlete B did not improve his aerobic power but did improve in his anaerobic power. And why Athlete A and C improved in both aerobic and anaerobic power. Also, discussion why individuals with cerebral palsy are expected to improve or not improve in their physical attributes as a result of the disability when going through a systematic physical training programme.
- Line 431 and throughout the manuscript, the study did not measure VO2max via respiratory gas measure bur rather determined the individuals’ VO2max via a regression equation from a progressive incremental exercise to exhaustion, thus the author should use the term estimated or predicted VO2max throughout the manuscript.
please see comments to authors
Author Response
Comments 1: [Throughout the manuscript. Please use decimal point (.) and not comma (,) for numeral values. Also, standardized the number of decimal points. For example in Table 3, sometimes you used 2 and sometimes 3 and sometimes no decimal point – please be consistent.] |
Response 1: Thank you for pointing this out. I/We agree with this comment. [The document was reviewed to ensure the consistent use of the decimal point (.) for all numerical values. A standardized decimal format was applied throughout the entire study.] “[updated text in the manuscript if necessary]” |
Comments 2: [Line 179-180. The authors replied to my initial comments: “Body composition was assessed using bioelectrical impedance (OMRON Healthcare Technology Co, Ltd., HBF-514C® scale, Kyoto, Japan). It is important to mention that this device has been used in previous studies as noted in the section on measurement of body composition.” The exact information that this reviewer is seeking is whether the equipment has previously been validated and the level of reliability of the equipment.] |
Response 2: Thank you for pointing this out. [The following information is integrated: "which has demonstrated validity (r = .942), reliability (ICC = .933–.993), and has been used in previous studies," supported by the bibliographic reference: Vasold KL, Parks AC, Phelan DML, Pontifex MB, Pivarnik JM. Reliability and Validity of Commercially Available Low-Cost Bioelectrical Impedance Analysis. International Journal of Sport Nutrition and Exercise Metabolism. 2019;29(4):406–410. https://doi.org/10.1123/ijsnem.2018-0283]” |
Comments 3: [Line 197. The regression equation was obtained from reference no 37. How was this regression equation obtained, i.e., from abled or disabled sample of participants? If disabled, is it from participants who are cerebral palsy. Basically, reviewer is seeking confirmation that the regression equation to determine VO2max is valid for use for the 3 participants of this study.] |
Response 3: Thank you for pointing this out. [Although this equation has been used in athletes without disabilities, previous studies show the need to use prediction or estimation models for VOâ‚‚max in sports that require high aerobic capacity40, Such as Paracycling in the T1 and T2 sport classes. In the discussion section, one of the study’s limitations is highlighted: 'Moreover, the equation used to estimate VOâ‚‚max has not been previously validated for athletes with cerebral palsy; therefore, the findings should be interpreted with caution.”] |
Comments 4: [Line 232. What type/model of cycle ergometer was sued during the 3 participants’ training session.] |
Response 4: Thank you for pointing this out. [The requested information is complemented with:“The participants performed all HIIT training sessions using a bicycle ergometer (Avantronic, Cyclus 2 ®, RBM elektronik-automation GmbH, Leipzig, Germany).”] |
Comments 5: [Line 239 and 240. “The training load ….. 3 minutes at 85% W, with a 3 min recovery …..60% W. Short HIIT at 100% W, with a …..at 20% W”. Authors needs to be very clear with their writing; for in the above sentence, the values of at 85% W, 60% W, 100% W and 20% W of what reference values. As it currently stand, the sentence is unclear.] |
Response 5: Thank you for pointing this out. [Table 1 presents the workload distribution during the HIIT sessions throughout the intervention. The training program consisted of eight sessions (long HIIT: four sessions; short HIIT: four sessions), and the training load was distributed according to the principles of individuality. Training intensity was determined using the results from the incremental test to program the long HIIT sessions and the WAnT_30 sec test to set the short HIIT sessions, with values measured in watts (W). Although the intensity was similar among the three athletes, the training volume differed based on the results of each test. The programmed volume for the long HIIT sessions consisted of 2 sets of 5 repetitions of a 3-minute work cycle, with a 3-minute rest between repetitions. In the long HIIT sessions, the intensity for the work intervals was set at 85% W (Athlete A: 123.3 W; Athlete B: 123.3 W; and Athlete C: 191.3 W) and at 60% W (Athlete A: 87 W; Athlete B: 87 W; and Athlete C: 135 W) for the recovery intervals. For the short HIIT sessions, the programmed volume consisted of 1 set of 10 repetitions of a 30-second work cycle, with 1 minute of rest between repetitions. The intensity for the short HIIT sessions was set at 100% W (Athlete A: 282.3 W; Athlete B: 272.5 W; and Athlete C: 473.7 W) during the work intervals, and 20% W (Athlete A: 56.5 W; Athlete B: 54.5 W; and Athlete C: 94.7 W) during the recovery intervals.]” |
Comments 6: [Table 1 is unclear. What is “Programmed” and what is “Media? Also, the values of the power output should be shown in both Watts as well as a % of relative. Line 268 to 172. This entire paragraph caused some confusion. Firstly, author stated that the 3 athletes followed a “structured training program”, and then said that the intensity was monitored through subjective perception (but this reviewer has the impression that training intensity was based on relative percentage of the athlete’s performances in the aerobic and anaerobic test, with values shown in Table 1). Then the authors mentioned that “a systematized training methodology was implemented and adjusted by the coach according to the individual characteristics of each participant”, again the reviewer have the impression that training load/intensity was already fixed according to the relative abilities of the participants (reviewer is referring to Table 1 again). Again, as I previous requested that the mean power output in the Wingate test achieved by the 3 individuals during each of their training session is displayed/shown to the readers – to provide readers as to what is actually being planned relative to what actually is performed by the said athlete. Further, the authors mentioned in their reply to my initial review: “It is essential to note that the intensity is similar for each athlete, but the volume of work mobilized in Watts changes, respecting the principle of individuality.” This is again confusing because the intensity is relative to each individual, but the amount or volume of work, to me at least is fixed – as stated by the from Line 238-242 where log HIIT consisted of “2 sets of 5 reps of 3 min with a 3 min rest”, and for short HIIT is “1 set of 10 reps of 30 s cycle with a 1 min rest between reps”.] |
Response 6: Thank you for pointing this out. [It is important to note that in Table 1, "Programmed" refers to the workload in watts that the athletes were expected to mobilize during each training session, and "Average" refers to the mean wattage actually achieved by the athletes during each session. The relative watt percentage (%) is expressed below: "Table 1 presents the workload distribution during the HIIT sessions throughout the intervention. The training program consisted of eight sessions (four long HIIT sessions and four short HIIT sessions), and the workload was distributed based on the principle of individuality. Training intensity was determined using the results from the incremental test to prescribe the long HIIT sessions and from the WAnT_30 sec test to establish the short HIIT sessions, both measured in watts. While training intensity was similar across all three athletes, the training volume differed according to the results of each test. The programmed volume for the long HIIT sessions consisted of 2 sets of 5 repetitions of a 3-minute cycle, with 3-minute rest intervals between repetitions. In these long HIIT sessions, the work intervals were performed at 85% W (Athlete A: 123.3 W; Athlete B: 123.3 W; and Athlete C: 191.3 W) and the rest intervals at 60% W (Athlete A: 87 W; Athlete B: 87 W; and Athlete C: 135 W). For the short HIIT sessions, the programmed volume consisted of 1 set of 10 repetitions of a 30-second cycle with 1-minute rest between repetitions. The intensity during the work intervals was set at 100% W (Athlete A: 282.3 W; Athlete B: 272.5 W; and Athlete C: 473.7 W), and at 20% W during the rest intervals (Athlete A: 56.5 W; Athlete B: 54.5 W; and Athlete C: 94.7 W). Furthermore, the workload/intensity was previously established as mentioned and is shown in Table 1 under the "Programmed" section. Regarding the suggestion to show readers the average power output based on the Wingate test across training sessions, it is important to clarify that Table 1 presents this data as the “Average” achieved during the long and short HIIT sessions. The Wingate test was used only at the beginning and end of the study. Finally, the training volume was fixed in terms of sets, repetitions, and recovery. However, the mobilized workload in watts differed for each athlete because it was programmed based on their individual aerobic and anaerobic test results. It is also important to note that the phrase “The workload intensity was initially monitored through subjective perception” has been removed prior to Table 2. |
Comments 7: [Line 253. Reviewer is concern about the stats in the paper. Firstly, when percentage changes is used in the calculation between pre and post-test for each test for each participant – why author need to include the paired samples significance test? Once you have the percentage changes – you compare the % change with the CV of the test. If the % change in the pre to post test is greater than 2 x CV, then it is deemed that a true or real change has occurred in that individual. If the % change in the pre to post test is less than 2 x CV, then this may or may not signify a true change, the value of the change is within the variability or ‘noise’ of the test. For example, in the changes in the estimated VO2max measure, athlete B clearly did not improve. And for the time-trial, Athlete B and C improves by a margin of 3.7%; is this small margin of improvements a true and real indication of actual performance improvements or again its merely within the variability or ‘noise’ of the time-trial test.] |
Response 7: Thank you for pointing this out. [For the statistical analysis, the paired samples significance test and the effect size (ES) were removed. Finally, the document was reviewed to ensure the correct use of the decimal point (.) for all numerical values, and a standardized decimal format was applied throughout the study. In the Results section, the variables that showed improvements are highlighted, considering that the percentage change must be greater than 2 × CV.] |
Comments 8: [Line 254. You should have only one CV value for each test. How was CV of each of the test calculated?.] |
Response 8: Thank you for pointing this out. It is obtained using the following formula for the coefficient of variation (CV = [Standard Deviation / Mean] * 100)]” |
Comments 9: [Figure 1 A and B is not useful at all. The 2 figures merely show the comparisons of the 3 athletes’ training HR and RPE, but alas, given the differences between the 3 athletes (as we have agreed) – the figures’ data are meaningless. It would have been better to show the HR, RPE and Power Output of each of the training session for each individual athletes across the 8 training sessions rather than averaging the session over a week.] |
Response 9: Thank you for pointing this out. [As suggested, heart rate (HR), rating of perceived exertion (RPE), and average power output for each session are presented, showing the individual workloads mobilized by each athlete throughout the 8 training sessions (see Table 1). On the other hand, Figure 1 A and B were removed, since the aforementioned table contextualizes the requested absolute and relative data.] |
Comments 10: [Table 2 and Table 3. I don’t understand what are the effects size (ES) comparing between which measure. To my knowledge, ES is used to compare between 2 measures. Inconsistency in Table 3 with sometimes the (,) and (.), please be consistent.] |
Response 10: Thank you for pointing this out. [For this study, the effect size (ES) was removed]” |
Comments 11: [Write max HR as HRmax.] |
Response 11: Thank you for pointing this out. [max HR" is written and replaced as "HRmax"] |
Comments 12: [The discussion section should primarily be about whether each athlete has performed the training that has been tasked upon him/her and whether there is actually ‘real’ improvements made by the athlete (this should be based on the percentage changes of performance relative to the CV of the test). For example, it will be appropriate and relevant to discuss why Athlete B did not improve his aerobic power but did improve in his anaerobic power. And why Athlete A and C improved in both aerobic and anaerobic power. Also, discussion why individuals with cerebral palsy are expected to improve or not improve in their physical attributes as a result of the disability when going through a systematic physical training programme.] |
Response 12: Thank you for pointing this out. [Athlete B did not show improvement in estimated VOâ‚‚max, unlike Athletes A and C. This may be due to the lack of sufficient implementation of long-type HIIT sessions, which could have helped the athlete better tolerate and delay the onset of fatigue during prolonged efforts, considering optimal volume, intensity, and recovery. This is evident (see Table 1) when comparing Athletes A and B, who belong to class T1. In the long HIIT sessions, Athlete B showed a lower average power output during the first two weeks. However, in the WAnT_30s test, this athlete displayed greater affinity for tests targeting the glycolytic and phosphagen energy systems, achieving a 15% improvement compared to the initial test. Athletes A and C showed improvements in both the estimated VOâ‚‚max and WAnT_30s tests. This may be attributed to their greater muscle mass, which contributes to enhanced recruitment of muscle fibers, and to the fact that the training program matched their needs and individual characteristics. Overall, due to the adaptation (in some tests and athletes) to the training program—as evidenced by data from the estimated VOâ‚‚max and WAnT_30s—a reduction in time trial performance was achieved for the specified distances..]” |
Comments 13: [Line 431 and throughout the manuscript, the study did not measure VO2max via respiratory gas measure bur rather determined the individuals’ VO2max via a regression equation from a progressive incremental exercise to exhaustion, thus the author should use the term estimated or predicted VO2max throughout the manuscript.] |
Response 13: Thank you for pointing this out. [The term estimated VOâ‚‚max is used throughout the manuscript.] |

Round 3
Reviewer 3 Report
Comments and Suggestions for Authors
Minor changes
For Table 1. Should “Athlete” rather than “Atleta”. Should be “Planned exercise intensity (W)” rather than “Programmed”. Should be “Performed mean exercise intensity (W)” rather than “Mean W(%)”.
Throughout the manuscript. Write out fully the word “max”.
Line 324. What is “W max”? Please write fully.
Author Response
Comments 1: [For Table 1. Should “Athlete” rather than “Atleta”. Should be “Planned exercise intensity (W)” rather than “Programmed”. Should be “Performed mean exercise intensity (W)” rather than “Mean W(%)”.] |
Response 1: Thank you for pointing this out. I/We agree with this comment. Therefore, I/we have….[In Table 1, the word 'Atleta' is replaced with 'Athlete'. In addition, 'Programmed' is changed to 'Planned exercise intensity (W)'. Finally, 'Performed mean exercise intensity (W)' will be used instead of 'Mean W(%)”.] “[updated text in the manuscript if necessary]” |
Comments 2: [Throughout the manuscript. Write out fully the word “max”.] |
Response 2: Thank you for pointing this out. [In the text, the abbreviation 'max' is written out in full as 'maximum watts (Wmax)' and 'maximum heart rate (HRmax) and “maximum oxygen consumption (VO2max)”.] |
Comments 3: [Line 324. What is “W max”? Please write fully.] |
Response 3: Thank you for pointing this out. [Improvements in maximum power output (Maximum Watts) during the WAnT_30s test after the HIIT program were observed in all participants: 31% (282.3 vs 370.4 W) in Athlete A, 15% (272.5 vs 312.6 W) in Athlete B, and 9% (473.7 vs 516.2 W) in Athlete C.]. |
